# An Investigation on Korean Adolescents’ Dietary Consumption: Focused on Sociodemographic Characteristics, Physical Health, and Mental Health

**DOI:** 10.3390/ijerph18189773

**Published:** 2021-09-16

**Authors:** Hui-Rang Yim, Hyun Ju Yun, Jee Hye Lee

**Affiliations:** 1Department of Food & Nutrition, College of Life and Health Sciences, Kyungsung University, 309 Suyeong-ro, Nam-gu, Daeyeon-dong, Busan 48434, Korea; hrlim@ks.ac.kr; 2Department of Food & Nutrition, College of Human Ecology, University of Ulsan, 93 Daehak-ro, Nam-gu, Ulsan 44610, Korea; goidcat@ulsan.ac.kr

**Keywords:** adolescents, dietary intake, sociodemographic characteristics, physical health, mental health

## Abstract

The promotion of healthful dietary intake in adolescents is an important public health concern, as obesity is on the rise among adolescents. The current study aimed to determine the association between sociodemographic characteristics and dietary intake (breakfast, vegetables, milk, fruits, fast food, carbohydrate beverages, and caffeine beverages) and to examine the influences of physical and mental health on dietary intake. A nationally representative sample of 187,622 adolescents who attended middle and high schools and participated in the Korean Youth Risk Behavior Survey (2015, 2017, and 2019) was used. The results showed a decrease in breakfast, vegetable, milk, and fruit consumption and an increase in the obesity rate from 2015 to 2019. There was a significant association between dietary intake and gender, economic status, BMI, and academic achievement. Multiple logistic regression analysis results revealed that the consumption of breakfast, vegetables, milk, and fruit were associated with greater odds of having enough sleep. Fast food intake was shown to be a significant risk factor for depression with an odds ratio of 1.204 (95% CI = 1.169–1.239), and carbonated beverage consumption was shown to be a significant risk factor for stress with an odds ratio of 1.042 (95% CI = 1.030–1.054). These results provide fundamental data that can be used to develop an effective nutrition intervention program or nutrition policy for adolescents or school food service programs.

## 1. Introduction

Adolescence is the transition period from childhood to adulthood, and it is a phase of physical and mental preparation for second maturity. Due to the rapid body growth that occurs in adolescence, this part of the life cycle has the highest nutritional requirements [1]. Healthy eating habits among children can prevent malnutrition, micronutrient deficiencies, obesity, or other diseases [2,3]. When adolescents do not have enough knowledge regarding their health and nutrition or are not fully aware of the importance of health, nutritional imbalances can easily occur, such as when individuals focus on taste in food choice [4]. Adolescence is a time when eating habits become fixed, and the correction of bad eating habits is difficult; therefore, eating habits in adolescents can affect health during adulthood [5].

Nutritional imbalance among Korean adolescents has become a problem along with the introduction of a westernized diet that includes instant foods and junk foods [6]. Eating fast food is already popular and has become a food culture among younger people, including children and adolescents [7]. Positive associations among an increase in fast food consumption, skipped breakfasts, and increased body mass index were reported among adolescents [8]. In addition, the deficient intake of fruits and vegetables has become a serious problem, although these foods are rich in micronutrients [9]. Rapid economic improvement has increased the consumption of processed foods and fast food; in addition, beverage consumption has also increased [10]. Food intake in adolescence has created an imbalance along with the introduction of foreign diets.

Studies on dietary intake among adolescents have focused on current intake status [11,12], the associations between physical activities and dietary behaviors [13,14], and the effects of nutrition education [15,16]. Some studies have researched the influence of physical activity on obesity [17,18]. Previous studies have examined the influence of physical activity on obesity [18] or the positive influence of physical activity and dietary intake on health [17]; however, there is relatively little cross-sectional research on the influence of physical and mental health on dietary behavior among adolescents. In addition, the association between stress and dietary intake [19,20] has been mostly conducted in randomly sampled groups of adolescents.

Adolescents are emotionally sensitive, and this instability may be associated with higher levels of unhealthy dietary practices. Several psychological factors and situations such as relationships with parents or friends that can be experienced in adolescence can interfere with healthy eating [21], and insufficient physical activities are also associated with unhealthy eating habits [22]. Few studies have been published on the investigation of the impact of physical and mental health on various types of food intake among Korean adolescents using national data, although understanding the determinants of dietary intake is important to effectively promote healthy dietary behavior among adolescents using the most recent big data. In particular, in this study, various types of food, including breakfast, fruit, and vegetable intake, and fast food, carbohydrate, and caffeine beverages, known as good and bad foods, respectively, were included. The research findings can help adolescents develop proper eating habits by describing the patterns and determinants of dietary intake, as well as to provide research evidence for nutrition experts or governors who develop nutrition guidelines and policies.

Thus, this study first identified the patterns of various types of dietary intake in Korean adolescents using the latest national representative survey data from 2015, 2017, and 2019, which represent all adolescents in Korea. Second, it explored the association between sociodemographic characteristics (gender, economic status, BMI, and academic achievement) and dietary intake. Third, this study examined the influences of physical and mental health on dietary intake.

## 2. Materials and Methods

### 2.1. Sample and Survey Administration

The present study was based on raw data from the Korean Youth Risk Behavior Survey (KYRBWS), which was conducted by the Ministry of Education (MOE), Ministry of Health and Welfare (MOHW), and Korea Centers for Disease Control and Prevention (KCDC) in 2015, 2017, and 2019. The KYRBWS collected cross-sectional data on a broad scope of health behaviors from a representative sample of middle- and high-school students. The survey used a stratified, clustered, multistage probability random sampling methodology that has been described previously [23], and it has been conducted every year since 2005 to understand Korean youth’s smoking, drinking, obesity, diet, exercise, and health conditions. The sampling design process was as follows: First, at the population stratification step, the population was divided by layer using stratification variables (17 regions throughout the country and 3 types of schools, including middle school, high school, and specialized school). Second, at the sample allocation steps, after distributing the sample size into 400 middle schools and 400 high schools, 5 middle schools and 5 high schools were allocated first for each of the 17 regions based on the proportional allocation method. Finally, at the sampling step, the stratification collection method was applied. For the first extraction, the school sample was selected. For the secondary extraction, 1 class was randomly selected by grade from the selected school sample. The random sampling can minimize response bias. Data from 2015, 2017, and 2019 were used for this study because the data needed in this study were only collected in those years. The sample size was 68,043 from 797 schools in 2015 (response rate 96.7%), 62,276 from 799 schools in 2017 (response rate 95.8%), and 57,303 from 800 schools in 2019 (response rate 95.3%); thus, a total of 187,622 participants were included in the final analysis.

### 2.2. Measurement

Table 1 describes the measures, questions, and analytic coding for each variable included in the analyses. Sociodemographic characteristics included gender, economic status, BMI, and academic achievement. Gender was classified into boys and girls. In terms of economic status and academic achievement, low and middle-low were converted to low, middle was converted to middle, and middle-high and high were converted to high. Regarding obesity, body mass index (BMI; kg/m^2^) was determined based on height and weight. According to the 2018 standard Korean society for the study of obesity guidelines, a BMI lower than 18.5 indicates underweight, a BMI between 18.5 and 23 indicates normal, a BMI between 23 and 25 indicates overweight, and a BMI higher than 25 indicates obesity.

Physical activity was assessed by asking ‘How many days have you experienced heart rate being higher than usual or have you done cardio exercise more than 60 min during the last 7 days?’ An eight-point Likert scale was used with the following response options: not in the last 7 days, 1 day, 2 days, 3 days, 4 days, 5 days, 6 days, or 7 days. Enough sleep was assessed by asking ‘Have you had enough sleep to recover from fatigue during the last 7 days?’ A five-point Likert scale was used with the following response options: more than enough, enough, okay, not enough, or not enough at all. In addition, this item was reverse coded. Stress was assessed by asking ‘How often do you feel stressed out?’ A five-point Likert scale was used with the following response options: very often, often, a little, not so much, or not at all. This item was also reverse coded. Experiences with depression and suicide attempts were assessed with yes or no questions, which were reverse coded.

Respondents were also asked about their daily dietary intake: breakfast, vegetables, milk, fruits, fast food, carbohydrate beverage, and caffeine beverage. The item assessing breakfast was ‘how many days in the last 7 days have you had breakfast?’ The response options were recoded as follows: no days (converted to not eaten); 1 day, 2 days, 3 days, 4 days, 5 days, 6 days, and 7 days (converted to eaten). The responses for the questions regarding how often vegetables, milk, fruits, and fast food were consumed were also recoded as follows: not in the last 7 days (converted to not eaten); once or twice a week, 3 or 4 times a week, 5 or 6 times a week, once every day, twice a day, and ≥3 times a day (converted to eaten). The carbohydrate and caffeine beverage data were divided into two groups: fewer or more than 2 cups/day. This standard of carbohydrate beverage was set based on the WHO (World Health Organization)’s recommendation that the daily intake of sugar is 10% of daily calories (the recommended daily calories for Korean adolescents are 2000–2700 kcal) and the fact that the average sugar content of carbohydrate beverages is 23–30 g per 8 ounces [24]. Regarding caffeine beverages, two cups was also set as the standard considering that the maximum daily caffeine intake is 125 mg for adolescents weighing 50 kg [25] and the average caffeine content of 35 drinks in Korea is 67.9 mg [26].

### 2.3. Statistical Analysis

Complex survey data analyses, including analyses of weight, stratification variables, and cluster variables, were conducted using SPSS Statistics 25.0 (SPSS Inc., IBM Corp., Armonk, NY, USA). The sample was weighted by statisticians in the Korean CDC (Weight = 1/fraction x 1/response rate x weight post-calibration rate). All data were examined using frequency analysis and expressed as a percentage of the mean ± SE. Then, chi-square analysis (a Rao-Scott χ^2^-test) was used to show trends of dietary intakes from 2015 to 2019 and to determine the association between sociodemographic characteristics (gender, economy status, BMI, and academic achievement) and subjects’ dietary intakes. Multiple logistic regression analysis was performed to examine the influence of physical and mental health on dietary intake. It was analyzed to obtain odds ratios (ORs) and 95% confidence intervals (CIs) adjusting for covariates. A *p*-value < 0.05 was considered statistically significant.

## 3. Results

Table 2 shows the general characteristics of adolescents of the subjects. The final analysis included 187,622 participants, including 35,204 (51.7%) boys and 32,839 (48.3%) girls in 2015, 31,624 (50.8%) boys and 30,652 (49.2%) girls in 2017, and 29,841 (52.1%) boys and 27,462 (47.9%) girls in 2019. Considering the school level, 2015 data included 10,786 adolescents in the 1st grade of middle school (15.9%), 11,442 adolescents in the 2nd grade of middle school (16.8%), 12,071 adolescents in the 3rd grade of middle school (17.7%), 11,122 adolescents in the 1st grade of high school (16.3%), 11,113 adolescents in the 2nd grade of high school (16.3%), and 11,509 adolescents in the 3rd grade of high school (16.9%). The 2017 year and 2019 year results also showed a similar distribution. Subjective economic status was considered ‘low,’ increasing from 359 (0.5%) in 2015 to 7341 (12.8%) in 2019, and subjective economic status was considered ‘high,’ decreasing from 65,352 (96.0%) in 2015 to 22,505 (39.3%) in 2019. BMI data showed that 7171 (11.1%), 8334 (14.1%), and 8580 (15.7%) adolescents were considered obese in 2015, 2017, and 2019, respectively. These results indicate that obesity has been on the rise.

The frequency of eating breakfast was found to decrease from 32,693 (48.0%) in 2015 to 22,267 (38.8%) in 2019 (*p* < 0.01). The frequency of vegetable intake also showed a decrease. A total of 20,332 (29.9%) and 13,804 (24.1%) participants reported ‘high vegetable intake’ in 2015 and 2019, respectively (*p* < 0.01). Regarding milk intake, it showed a decrease from 7650 (11.2%) in 2015 to 4979 (8.7%) in 2019 (*p* < 0.01). The fruit intake showed a decrease as follows (*p* < 0.01): the number of adolescents reporting ‘no intake’ increased from 6209 (9.1%) in 2015 to 6234 (10.9%) in 2019. Fast food showed an increase as follows: the number of adolescents reporting ‘high intake’ increased from 298 (0.5%) in 2015 to 437 (0.8%) in 2019 (*p* < 0.01). Regarding beverage consumption, carbonated beverage and caffeinated beverage consumption showed an increase. The intake of more than two carbonated beverages was reported by 19,151 (28.1%) adolescents in 2015 and 21,156 (36.9%) in 2019 (*p* < 0.01). The intake of more than two carbonated caffeinated beverages was reported by 2192 (3.2%) adolescents in 2015 and 6799 (11.9%) in 2019 (*p* < 0.01).

Table 3 shows the descriptive analysis results of physical and mental health. Regarding physical health, the mean ‘physical activity’ was 2.96 in 2015, 2.93 in 2017, and 3.03 in 2019. The mean enough sleep decreased from 2.88 in 2015 to 2.67 in 2019. Regarding mental health, the mean ‘stress’ increased from 3.19 in 2015 to 3.28 in 2019. Moreover, the frequency of individuals reporting depression increased from 15,894 (23.4%) in 2015 to 16,028 (28.0%) in 2019, and the frequency of suicide attempts increased as follows: 1662 (2.4%) in 2015, 1634 (2.6%) in 2017, and 1731 (3.0%) in 2019.

Table 4 shows the associations of dietary intake with gender. Boys showed more frequent vegetable (96.4% vs. 95.6%) and milk consumption (87.9% vs. 81.1%) than girls, whereas girls showed a higher breakfast intake (83.1% vs. 82.5%) and fruit consumption (91.1% vs. 89.3%) than boys did. Fast food (79.6% vs. 77.6%) and beverage consumption, including carbonated beverages (39.8% vs. 24.9%) and caffeinated beverages (8.2% vs. 6.7%), were also higher among boys than girls.

The results (Table 5) showed that the higher the economic status, the higher the breakfast (high: 84.3%, middle: 81.2%, and low: 78.2%), vegetable (high: 96.3%, middle: 95.8%, and low: 94.3%), milk (high 85.3%, middle 84.0%, and low 81.9%), and fruit intake (high 91.5%, middle 89.3%, and low 83.3%). On the other hand, under lower economic conditions, fast food (low 79.5%, middle 81.2%, and high 77.3%), carbonated beverage (low 37.3%, middle 34.4%, and high 31.2%), and caffeinate beverage intake increased (low 11.1%, middle 8.9%, and high 6.3%).

As Table 6 shows, regarding the association of dietary intake with BMI (underweight, normal range, overweight, and obese), the ‘obese’ group consumed the most vegetables (96.6%), carbonated beverages (34.3%), and caffeine beverages (9.0%); this group also reported the highest frequencies of not consuming breakfast (17.9%) and fruit (11.7%). The ‘normal’ group consumed the most fast food (79.1%). The ‘overweight’ group had the most milk consumption (85.2%).

The results of the analysis on the associations of dietary intake with the level of academic achievement (Table 7) showed that the higher the academic achievement was, the higher the breakfast (85.3%), milk (86.9%), and fruit (92.7%). On the other hand, the results showed that the lower the academic achievement was, the higher the intake of fast food (81.3%), carbohydrate beverages (43.5%), and caffeinated beverages (10.9%).

Table 8 and Table 9 represent the results of a logistic regression with dietary intake (breakfast, vegetable, milk, fruit, fast food, carbonated beverage, and caffeinated beverage) as the dependent variable. The results are presented as odds ratios with 95% confidence intervals. Consuming all of the dietary intake was associated with physical activity. The odds ratio for breakfast was 1.045 (95% CI = 1.038–1.051); vegetable was 1.154 (95% CI = 1.137–1.171); milk was 1.154 (95% CI = 1.145–1.163); fruit was 1.091 (95% CI = 1.081–1.101); fast food was 1.011 (95% CI = 1.005–1.016); carbonated beverage was 1.076 (95% CI = 1.070–1.081); and caffeinated beverage was 1.068 (95% CI = 1.059–1.077). Enough sleep was associated with consumption of breakfast (OR = 1.082, 95% CI = 1.069–1.095), vegetables (OR = 1.099, 95% CI = 1.071–1.127), milk (OR = 1.081 95% CI = 1.067–1.095), and fruit (OR = 1.077, 95% CI = 1.061–1.094). However, fast food (OR = 0.924, 95% CI = 0.914–0.934), carbonated beverage (OR = 0.951, 95% CI = 0.942–0.961), and caffeinated beverage (OR = 0.818, 95% CI = 0.802–0.834) were shown to be significant risk factors for enough sleep.

Regarding mental health, adolescents’ stress was associated with the consumption of breakfast (OR = 0.959, 95% CI = 0.945–0.973); vegetables (stress OR = 0.885, 95% CI = 0.855–0.910), milk (OR = 0.908, 95% CI = 0.894–0.922), fruit (OR = 0.852, 95% CI = 0.836–0.868), carbonated beverage (OR = 1.042, 95% CI = 1.030–1.054), and caffeinated beverage (OR = 1.209, 95% CI = 1.181–1.238). Depression was associated with the consumption of breakfast (depression OR = 0.889, 95% CI = 0.863–0.915), vegetables (depression OR = 0.932, 95% CI = 0.884–0.982), carbonated beverage (OR = 1.173, 95% CI = 1.144–1.202), and caffeinated beverage (OR = 1.445, 95% CI = 1.386–1.505). Suicide attempts were associated with consumption of breakfast (OR = 0.761, 95% CI = 0.711–0.814), fruit (OR = 0.696, 95% CI = 0.644–0.753), carbonated beverage (OR = 1.308, 95% CI = 1.231–1.389), and caffeinated beverage (OR = 1.842, 95% CI = 1.702–1.994). Additionally, the increased consumption of fast food was associated with greater odds of becoming depressed (OR = 1.204, 95% CI = 1.169–1.239).

## 4. Discussion

The current study investigated the association between sociodemographic characteristics (gender, economic status, BMI, and academic achievement) and six types of food intake and determined the influences of physical and mental health on dietary intake. The results showed that gender was associated with adolescents’ dietary intake. Female students showed more frequent breakfast and fruit consumption, whereas male students showed a higher consumption of vegetables and fast food. Regarding beverage consumption, female students had more caffeine beverages, whereas males drank more milk and carbohydrate beverages. Our results confirmed previous evidence partially. According to a study researching 2599 9~14-year-old children and adolescents from 2002 to 2011 [27], the amount of sugar intake from fruit is higher among boys than girls. In general, women tend to have more health-promoting eating behaviors than men [28]. Men give lower priority to health and consider taste or convenience in their food choices, so men consume fewer high-fiber foods, eat fewer fruits and vegetables and low-fat foods, and consume more soft drinks [29,30].

Regarding the analysis of economic status, from 2015 to 2019, the proportion of Korean adolescents who perceived their economic level to be low increased from 0.5% to 12.8%, and adolescents who perceived their economic level to be high decreased from 96% to 39.3%. This indicates that the proportion of students who perceived that their subjective economic condition was bad increased. The overall results showed that the worse the perceived economic condition was, the worse the dietary intake. Specifically, groups who perceive their economic status as low had a lower consumption of breakfast, vegetable, and milk but a higher consumption of fast food, and carbonated and caffeine beverages. Our results are also in accordance with previous evidence examining the significant association between economic status and dietary intake [31,32,33,34]. Jang et al. [32] examined the association between economic level and eating habits in Korean sixth-grade school children, where girls in the lower economic group showed a higher frequency of instant noodle consumption and skipping breakfast and a lower frequency of fruits, fish, and shell consumption. The reasons for this were mentioned in previous studies. Poor people tend to consume high-energy and high-fat foods, processed grains, and foods containing sugar, and buy low-cost foods because of limited budgets [35,36]. In addition, it is hard for these individuals to buy and prepare healthy meals because of time constraints of work commitments [36].

Regarding BMI, the results of the current study showed that obesity among Korean adolescents is on the rise (11.1% in 2015, 14.1% in 2017, and 15.8% in 2019), and overweight also has increased (11.7% in 2015, 12.2% in 2017, and 12.3% in 2019). Similarly, the results showed a gradual increase from 2015 to 2019 in carbohydrate and caffeine drink consumption among Korean adolescents. The percentage of drinking more than two carbohydrate beverages per day was 28.1% in 2015, 33.5% in 2017, and 36.9% in 2019, and more than two caffeine beverages per day was 3.2% in 2015, 7.8% in 2017, and 11.9% in 2019.

The findings showed that the obese group consumed carbohydrates and caffeine beverages more frequently and consumed less breakfast and vegetables. Numerous studies agreed with the associations between dietary intake and obesity [37]. This finding was consistent with previous studies determining the significant association between skipping breakfast and an increased likelihood of obesity [38] and the association between the intake of vegetables and fruits and an increased likelihood of obesity [39]. The intake of milk has an inverse relationship with obesity because calcium in the form of dairy products exerts an influence on weakening fat deposition [40]. The positive association between BMI and the frequency of carbohydrate beverages in the current study was similar to that in the study of Lee et al. [41]; carbohydrates from beverages accounted for the largest portion of carbohydrate consumption in processed foods among adolescents. The carbohydrate consumption from processed food in 2013 was 58.7 g, which exceeded the recommended level of the WHO (50 g) [41]. The 2015 WHO guidelines recommend that the sugar intake of children and adults should be less than 10% of the total daily energy intake. Shim [42] pointed out that 32.1 percent of Koreans have more than 20 g of sugar in their drinks, and one out of every three Koreans has an overconsumption of drinks [42]. Most carbonated beverages are sweetened drinks [43], so the excessive intake of sugars by drinking carbonated beverages can result in an imbalance in adolescents’ nutrition. The consumption of carbonated beverages can cause calcium loss due to an imbalance between calcium and phosphorus [44].

However, the results of fast food consumption in this current study showed that among the four groups, the normal group showed the highest intake, which is not consistent with most previous studies. The reason for this may be a unique situation of Korean students. Many Korean students spend much time at school or private institutions outside the house due to the heavy academic burden; thus, fast food consumption is easy because it satisfies students’ flavor-biased food selection and limited pocket money. Furthermore, fast food restaurants have convenient access [45]. These circumstances may encourage the consumption of normal group students.

The results of the current study supported the association between academic achievement and dietary intake. In detail, the findings showed that students with high academic performance showed a higher frequency of breakfast, vegetables, milk, and fruit consumption and lower carbohydrate beverage consumption. This finding is consistent with numerous studies that determined the association between school achievement and dietary intake. The five studies in the systemic review paper determined the associations between dietary intake and academic achievement (GPA) among university/college students’ studies, whereby students had regular breakfast meals and a higher consumption of fruit [46]. As Korean middle school students’ sleep hours and the average study time per day increased, the regularity of diet increased [47]. The higher the academic achievement, the more nutrients such as energy, protein, phosphorus, potassium, zinc, unsaturated fatty acid, and n-6 fatty acid the students were taking based on the study that examined the relationship between academic achievement and nutrient intake status [48]. In particular, a significant positive association between better school grades and daily breakfast consumption was found [49]. The reason is that tryptophan intake at breakfast helps serotonin and melatonin synthesis, resulting in a reduction in stress and promotion of better mental health [50].

The results showed that students with low academic performance drink caffeine beverages most. There is little research that explains why these factors are associated. Korean high-school students drink high-caffeine energy drinks to fight off sleepiness even though they know its side-effects [51]; it is assumed that they drink more to study. Adolescents’ caffeine drink consumption of more than the recommended amounts increased from 2015 (3.2%) to 2019 (11.9%). As the data show, overall, students drink much caffeine. Previous studies have emphasized the risk of caffeine consumption in adolescents. For adults, less than 100 mg/day of caffeine intake is effective in reducing fatigue and awakening; however, ≥1000 mg/day intake can cause consciousness disorders such as anxiety or insomnia, and ≥1500 mg/day intake can cause gastrointestinal disorders, arrhythmia, etc. [52]. Long-term caffeine intake promotes calcium malabsorption and calcium emission through urine, resulting in reduced bone density [53]. Adolescents are more sensitive because their bodies are smaller than those of adults, and the central nervous system is growing [54].

Unlike South Korea, in countries that are aware of these problems, the sale of a high-caffeine energy drink is restricted. For example, the sale of a high-caffeine energy drink to people under the age of 15 is prohibited in Sweden [55]. However, in Korea, buying energy drinks is easy at convenience stores, and advertisements using terms such as ‘for examination’ induce adolescents to purchase [53]. There is a lack of research about intervention programs to reduce high caffeine consumption, and interventions have not been actively conducted at school [51]. Thus, efforts for effective intervention programs to intake high-caffeine beverages should be made to protect the health of adolescents. An examination of the association between caffeine beverage intake and academic achievement is meaningful to research as many Korean students study a lot compared to other OECD (Organisation for Economic Co-operation and Development) averages [56].

The results of this study confirmed the effects of adolescents’ physical health on dietary intake overall. The more physical activity and enough sleep from good sleep, the more breakfast, vegetable, milk, and fruit were consumed. Previous studies have suggested that physically active people tend to eat healthy diets compared to sedentary people. Compared to sedentary adults, physically active individuals consume more fruits and vegetables and have higher intakes of fiber and calcium [57,58], which can have a positive influence on appetite and energy intake [59]. According to a study on the perception of eating habits according to the presence or absence of exercise in Korean college students [60], the exercise group showed less unbalanced dietary behavior and a higher consumption of green and yellow vegetables, seaweed, milk, and dairy products than the nonexercise group [60]. Additionally, the results of a study of 300 adolescents aged 12~18 found a positive effect of regular exercise on diet quality [61]. Physical activity influences dietary behavior by improving satiety, changing macronutrient preference, and modulating hedonic responses to foods [62,63,64,65]. Additionally, individuals with higher habitual physical activity levels than inactive individuals have a sensitive appetite regulation system through better compensatory adjustment of the energy content and density of food.

Overall, the results showed that mental health affects dietary intake among Korean adolescents. This finding was consistent with previous studies supporting the effect of stress on dietary choice or preference [66,67], the association between depression and dietary intake [68,69], and the association between suicide risk and dietary behavior [70]. Negative emotions can cause eating low-quality foods such as sweet snacks and fast food [71] and consuming fewer fruit, vegetables, whole grains, poultry, fish, and reduced-fat dairy products [72]. Stressed emotional eaters consume more sweet, high-fat foods and more energy-dense foods than unstressed and nonemotional eaters do [71]. More severe depression is associated with a higher total energy intake from saturated fat and sugars [73] and higher sweet food consumption [74]. According to the study of Michael [70], suicidal thoughts and attempted suicide were associated with skipping breakfast on all 7 days and consuming soda or pop [70]. People who are depressed tend to eat more junk food and gain more calories [75]. In the long run, unhealthy intake contributes to weight gain and increasing obesity. Depressed individuals tend to consume more calories and greater amounts of junk food.

This study has academic and practical contributions. First, the findings of this current study reflect the change in the dietary pattern of Korean adolescents by analyzing recent data. Adolescents are sensitive emotionally. This unstable status can be associated with higher levels of unhealthy dietary practices, so a study on the dietary intake of adolescents can provide research evidence for the development of a nutrition intervention program or nutrition policy. Moreover, as obese adolescents are on the rise, the promotion of healthful dietary intake in adolescents is already an important public health concern [76,77]. Based on the recent national data of Korean adolescents, identifying their food intake patterns and associations between sociodemographic characteristics and dietary intake are meaningful. Second, there is a lack of studies exploring the effect of physical and mental health on dietary intake. Numerous studies have determined the strong effect of eating habits on physical and mental health and emphasized the importance of healthy eating behaviors; however, comparatively limited studies have examined the reverse relationship.

At schools, nutrition intervention programs are subdivided according to various sociodemographic characteristics for Korean adolescents’ healthy dietary intake habits. For example, intervention programs should provide accurate information on the possible problems of indiscriminate caffeine consumption. In addition, adolescents’ physical and mental health is considered a critical influencing variable on dietary intake, and a nutrition intervention program that combines exercise and mental counseling can maximize the effect.

This current study has some limitations. First, the KYRBWS data included only six types of food intake (breakfast, fruit and vegetable, fast food, and two types of beverages) based on self-reported questions, which poses the probability of underreporting due to the unawareness of one’s behavior or social desirability [66]. In addition, in this current study, potential confounding factors including socioeconomic status (SES) were not considered and controlled in the multiple logistic regression model. Future studies could focus efforts on investigating diverse types of food group intake using objective data, taking into account the confounding factors. Second, the questionnaire asking about the frequency of food intake, not the actual serving amount of food consumed, was estimated. Third, the past week’s food intake was limited to reflecting past eating trends. Thus, a long-term follow-up cohort study analyzing actual amounts of food consumed based on the long-term period’s food intake is recommended as adolescents become older in the future. Finally, although the results confirmed the impacts of physical and mental health on dietary intakes, small R^2^ values may weaken the independent variables as good explanatory variables in the regression equation. Despite these limitations, the current study has strengths. This study is based on a nationally representative sample of Korean adolescents, so it may provide sufficient power and valid generalization.

## 5. Conclusions

Our study suggested evidence that the dietary intake of Korean adolescents changed significantly from 2015 to 2019. It confirmed the significant association between sociodemographic characteristics and seven types of dietary intake, and the positive influences of physical and mental health on dietary intake. The patterns of dietary intake from 2015 to 2019 showed a significant difference in food consumption between boys and girls. The better the economic status and academic achievement, the higher the breakfast consumption and lower the fast food and carbohydrate beverage consumption. Physical health, including physical activity and enough sleep, significantly positively affected breakfast, vegetable, milk, and fruit consumption. However, mental health, including stress, depression, and suicide attempts, showed a negative association with breakfast, vegetable, and fruit consumption. Nutrition practitioners at middle and high schools can consider the importance of physical and mental health in dietary behavior and apply it to develop a nutrition intervention program combined with exercise time and psychological counseling.

## Figures and Tables

**Table 1 ijerph-18-09773-t001:** Question and analytic coding for independent variables.

Variable		Question	Analytic Coding
Gender		What is your gender?	(1) Boys (2) Girls
Economy status		What is your family’s economy status?	(1) Low (2) Middle (3) High
BMI		kg/m^2^	(1) Underweight (2) Normal (3) Overweight (4) Obesity
Academic achievement		During the last 12 months, how are your academic grades?	(1) Low (2) Middle (3) High
Physical health	Physical activity	How many days have you experienced heart rate being higher than usual or have you done cardio exercise more than 60 min during the last 7 days?	(1) Not in the last 7 days (2) 1 day (3) 2 days (4) 3 days (5) 4 days (6) 5 days (7) 6 days (8) 7 days
	Enough sleep	Have you had enough sleep to recover from fatigue during the last 7 days?	(1) Not enough at all (2) Not enough (3) Okay (4) Enough (5) Very enough
Mental health	Stress	How often do you feel stressed out?	(1) Not at all (2) Not so much (3) A little (4) Often (5) Very often
	Depression experience	During the last 12 months, have you ever felt sad or desperate enough to quit your daily activities for the entire 2 weeks?	(1) No (2) Yes
	Suicide attempts	During the last 12 months, have you ever seriously considered suicide?	(1) No (2) Yes

**Table 2 ijerph-18-09773-t002:** General characteristics of adolescents (*N* = 187,622).

Characteristics	Items	2015	2017	2019	*p*-Value (χ^2^)
		*N* (%)	*N* (%)	*N* (%)	
Gender	Boys	35,204 (51.7)	31,624 (50.8)	29,841 (52.1)	0.995 (0.380)
	Girls	32,839 (48.3)	30,652 (49.2)	27,462 (47.9)	
Grade	Middle school 1st	10,786 (15.9)	10,189 (16.4)	9738 (17.0)	0.001 (357.785)
	Middle school 2nd	11,442 (16.8)	10,377 (16.7)	9665 (16.9)	
	Middle school 3rd	12,071 (17.7)	10,319 (16.6)	9981 (17.4)	
	High school 1st	11,122 (16.3)	10,165 (16.3)	9273 (16.2)	
	High school 2nd	11,113 (16.3)	10,800 (17.3)	9044 (15.8)	
	High school 3rd	11,509 (16.9)	10,426 (16.7)	9602 (16.8)	
Economy status	Low	359 (0.5)	8892 (14.3)	7341 (12.8)	0.001 (0.19)
	Middle	2332 (3.5)	28,582 (45.9)	27,457 (47.9)	
	High	65,352 (96.0)	24,802 (39.8)	22,505 (39.3)	
BMI	Underweight	15,013 (23.2)	12,557 (21.3)	11,541 (21.2)	0.001 (649.696)
	Normal range	34,821 (54.0)	30,959 (52.4)	27,639 (50.8)	
	Overweight	7538 (11.7)	7184 (12.2)	6671 (12.3)	
	Obese	7171 (11.1)	8334 (14.1)	8580 (15.7)	
Breakfast	No intake	10,076 (14.8)	10,946 (17.6)	11,444 (20.0)	0.001 (1275.892)
	Low (1~2/week)	9044 (13.3)	8712 (14.0)	9103 (15.9)	
	Middle (3~5/week)	16,230 (23.9)	15,014 (24.1)	14,489 (25.3)	
	High (6~7/week)	32,693 (48.0)	27,604 (44.3)	22,267 (38.8)	
Vegetable	No intake	2528 (3.7)	2597 (4.1)	2331 (4.1)	0.001 (577.729)
	Low (1~4/week)	27,304 (40.1)	24,779 (39.8)	24,756 (43.2)	
	Middle (5~7/week)	17,879 (26.3)	16,982 (27.3)	16,412 (28.6)	
	High (≥2/day)	20,332 (29.9)	17,918 (28.8)	13,804 (24.1)	
Milk	No intake	10,231 (15.1)	9047 (14.5)	8890 (15.5)	0.001 (576.749)
	Low (1~4/week)	28,528 (41.9)	28,532 (45.8)	27,209 (47.5)	
	Middle (5~7/week)	21,634 (31.8)	18,950 (30.5)	16,225 (28.3)	
	High (≥2/day)	7650 (11.2)	5746 (9.2)	4979 (8.7)	
Fruit	No intake	6209 (9.1)	6242 (10.0)	6234 (10.9)	0.001 (200.743)
	Low (1~4/week)	39,102 (57.5)	35,264 (56.6)	33,332 (58.1)	
	Middle (5~7/week)	15,635 (23.0)	14,420 (23.2)	12,598 (22.0)	
	High (≥2/day)	7097 (10.4)	6348 (10.2)	5139 (9.0)	
Fast food	No intake	17,718 (26.0)	12,646 (20.3)	10,517 (18.3)	0.001 (1633.073)
	Low (1~4/week)	48,454 (71.2)	47,216 (75.8)	43,647 (76.2)	
	Middle (5~7/week)	1573 (2.3)	2084 (3.4)	2702 (4.7)	
	High (≥2/day)	298 (0.5)	330 (0.5)	437 (0.8)	
Carbonated beverage	≦2 drinks	48,892 (71.9)	41,423 (66.5)	36,147 (63.1)	0.001 (1087.498)
	>2 drinks	19,151 (28.1)	20,853 (33.5)	21,156 (36.9)	
Caffeinated beverage	≦2 drinks	65,851 (96.8)	57,404 (92.2)	50,504 (88.1)	0.001 (3616.815)
	>2 drinks	2192 (3.2)	4872 (7.8)	6799 (11.9)	
Total		68,043(100)	62,276 (100)	57,303 (100)	

**Table 3 ijerph-18-09773-t003:** Descriptive analysis results of physical and mental health.

Characteristics			Mean (SD)		
			2015	2017	2019
Physical health	Physical activity		2.96 (2.097)	2.93 (2.092)	3.03 (2.129)
	enough sleep		2.88 (1.123)	2.78 (1.138)	2.67 (1.122)
Mental health	Stress		3.19 (0.948)	3.23 (0.979)	3.28 (0.992)
	Depression experience	Frequency, *N* (%)			
		Never	52,149 (76.6)	46,664 (74.9)	41,275 (72.0)
		Experience	15,894 (23.4)	15,612 (25.1)	16,028 (28.0)
		Total	68,043 (100)	62,276 (100)	57,303 (100)
	Suicide attempts	Never	66,381 (97.6)	60,642 (97.4)	55,572 (97.0)
		Attempts	1662 (2.4)	1634 (2.6)	1731 (3.0)
		Total	68,043 (100)	62,276 (100)	57,303 (100)

**Table 4 ijerph-18-09773-t004:** Associations of dietary intakes with gender.

	Dietary Intakes	Breakfast		Vegetable		Milk		Fruit		Fast Food		Carbonated Beverage		Caffeinated Beverage	
Gender		No Intake	Intake	No Intake	Intake	No Intake	Intake	No Intake	Intake	No Intake	Intake	<2 Cups	≥2 Cups	<2 Cups	≥2 Cups
Boys		17.5%	82.5%	3.6%	96.4%	12.1%	87.9%	10.7%	89.3%	20.4%	79.6%	60.2%	39.8%	91.8%	8.2%
Girls		16.9%	83.1%	4.4%	95.6%	18.9%	81.1%	8.9%	91.1%	22.4%	77.6%	75.1%	24.9%	93.3%	6.7%
*p*(F)		0.004(8.306)		0.001(62.432)		0.001(839.611)		0.001(122.896)		0.001(68.649)		0.001(2440.094)		0.001(56.581)	

**Table 5 ijerph-18-09773-t005:** Associations of dietary intake with the level of economic status.

	Dietary Intakes	Breakfast		Vegetable		Milk		Fruit		Fast Food		Carbonated Beverage		Caffeinated Beverage	
Economy Status		No Intake	Intake	No Intake	Intake	No Intake	Intake	No Intake	Intake	No Intake	Intake	<2 Cups	≥2 Cups	<2 Cups	≥2 Cups
Low		21.8%	78.2%	5.7%	94.3%	18.1%	81.9%	16.2%	83.8%	20.5%	79.5%	62.7%	37.3%	88.9%	11.1%
Middle		18.8%	81.2%	4.2%	95.8%	16.0%	84.0%	10.7%	89.3%	18.8%	81.2%	65.6%	34.4%	91.1%	8.9%
High		15.7%	84.3%	3.7%	96.3%	14.7%	85.3%	8.5%	91.5%	22.7%	77.3%	68.8%	31.2%	93.7%	6.3%
p(F)		0.001 (246.934)		0.001 (67.630)		0.001 (56.954)		0.001 (423.427)		0.001 (149.387)		0.001 (124.953)		0.001 (306.922)	

**Table 6 ijerph-18-09773-t006:** Associations of dietary intakes with BMI.

	Dietary Intakes	Breakfast		Vegetable		Milk		Fruit		Fast Food		Carbonated Beverage		Caffeinated Beverage	
BMI		No Intake	Intake	No Intake	Intake	No Intake	Intake	No Intake	Intake	No Intake	Intake	<2 Cups	≥2 Cups	<2 Cups	≥2 Cups
Underweight		16.7%	83.3%	4.3%	95.7%	15.3%	84.7%	8.9%	91.1%	21.4%	78.6%	66.7%	33.1%	94.1%	5.9%
Normal range		17.1%	82.9%	3.9%	96.1%	15.6%	84.4%	9.4%	90.6%	20.9%	79.1%	68.3%	31.7%	92.7%	7.3%
Overweight		17.1%	82.9%	3.7%	96.3%	14.8%	85.2%	10.1%	89.9%	21.9%	78.1%	68.1%	31.9%	92.1%	7.9%
Obese		17.9%	82.1%	3.4%	96.6%	15.2%	84.8%	11.7%	88.3%	22.4%	77.6%	65.7%	34.3%	91.0%	9.0%
*p*(F)		0.001 (5.354)		0.001 (10.464)		0.046 (2.676)		0.001 (48.722)		0.001 (10.102)		0.001 (23.592)		0.001 (72.972)	

**Table 7 ijerph-18-09773-t007:** Associations of dietary intake with the level of academic achievement.

	Dietary Intakes	Breakfast		Vegetable		Milk		Fruit		Fast Food		Carbonated Beverage		Caffeinated Beverage	
Academic Achievement		No Intake	Intake	No Intake	Intake	No Intake	Intake	No Intake	Intake	No Intake	Intake	<2 Cups	≥2 Cups	<2 Cups	≥2 Cups
Low		26.0%	74.0%	7.3%	92.7%	18.8%	81.2%	17.6%	82.4%	18.7%	81.3%	56.5%	43.5%	89.1%	10.9%
Middle		16.8%	83.2%	3.7%	96.3%	15.4%	84.6%	9.5%	90.5%	21.5%	78.5%	68.1%	31.9%	93.0%	7.0%
High		14.7%	85.3%	3.9%	96.1%	13.1%	86.9%	7.3%	92.7%	22.1%	77.9%	68.6%	31.4%	90.9%	9.1%
*p*(F)		0.001 (390.589)		0.001 (197.375)		0.001 (99.529)		0.001 (560.192)		0.001 (33.831)		0.001 (359.175)		0.001 (184.932)	

**Table 8 ijerph-18-09773-t008:** Odds ratios for dietary intake according to physical and mental health.

	D.V.	Breakfast ^a^		Vegetable ^b^		Milk ^c^		Fruit ^d^	
I.V.		OR (95% CI)	*p*-Value	OR (95% CI)	*p*-Value	OR (95% CI)	*p*-Value	OR (95% CI)	*p*-Value
Physical health	Physical activity	1.045 (1.038~1.051)	0.001	1.154 (1.137~1.171)	0.001	1.154 (1.145~1.163)	0.001	1.091 (1.081~1.101)	0.001
	Enough sleep	1.082 (1.069~1.095)	0.001	1.099 (1.071~1.127)	0.001	1.081 (1.067~1.095)	0.001	1.077 (1.061~1.094)	0.001
Mental health	Stress	0.959 (0.945~0.973)	0.001	0.885 (0.855~0.910)	0.001	0.908 (0.894~0.922)	0.001	0.852 (0.836~0.868)	0.001
	Depression	0.889 (0.863~0.915)	0.001	0.932 (0.884~0.982)	0.001	1.006 (0.975~1.038)	0.723	0.984 (0.948~1.021)	0.391
	Suicide attempts	0.761 (0.711~0.814)	0.001	0.552 (0.496~0.615)	0.001	0.940 (0.871~1.014)	0.110	0.696 (0.644~0.753)	0.001

Note. D.V. = dependent variable; I.V. = independent variable. ^a^ Nagelkerke’s R^2^ was 0.008 (Cox and Snell’s R^2^ was 0.005); ^b^ Nagelkerke’s R^2^ was 0.019 (Cox and Snell’s R^2^ was 0.005); ^c^ Nagelkerke’s R^2^ was 0.023 (Cox and Snell’s R^2^ was 0.013); ^d^ Nagelkerke’s R^2^ was 0.016 (Cox and Snell’s R^2^ was 0.007); OR = odds ratio; CI = confidence interval.

**Table 9 ijerph-18-09773-t009:** Odds ratios for dietary intake according to physical and mental health (continued).

	D.V.	Fast Food ^e^		Carbonated Beverage ^f^		Caffeinated Beverage ^g^	
I.V.		OR (95% CI)	*p*-Value	OR (95% CI)	*p*-Value	OR (95% CI)	*p*-Value
Physical health	Physical activity	1.011 (1.005~1.016)	0.001	1.076 (1.070~1.081)	0.001	1.068 (1.059~1.077)	0.001
	Enough sleep	0.924 (0.914~0.934)	0.001	0.951 (0.942~0.961)	0.001	0.818 (0.802~0.834)	0.001
Mental health	Stress	1.009 (0.996~1.023)	0.155	1.042 (1.030~1.054)	0.001	1.209 (1.181~1.238)	0.001
	Depression	1.204 (1.169~1.239)	0.001	1.173 (1.144~1.202)	0.001	1.445 (1.386~1.505)	0.001
	Suicide attempts	0.905 (0.843~0.972)	0.006	1.308 (1.231~1.389)	0.001	1.842 (1.702~1.994)	0.001

Note. D.V. = dependent variable; I.V. = independent variable. ^e^ Nagelkerke’s R^2^ was 0.005 (Cox and Snell’s R^2^ was 0.003); ^f^ Nagelkerke’s R^2^ was 0.012 (Cox and Snell’s R^2^ was 0.009); ^g^ Nagelkerke’s R^2^ was 0.039 (Cox and Snell’s R^2^ was 0.016); OR = odds ratio; CI = confidence interval.

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
