# Peer review of "An Investigation on Korean Adolescents’ Dietary Consumption: Focused on Sociodemographic Characteristics, Physical Health, and Mental Health"

_ijerph, 2021, doi:10.3390/ijerph18189773_

Round 1
Reviewer 1 Report
There is a lack of studies exploring the effect of physical and mental health on dietary intake. This cross-sectional study investigated the association between sociodemographic characteristics (gender, economic status, BMI, and academic achievement) and 6 types of food intake and determined the influences of physical and mental health on dietary intake using 2015, 2017 and 2019 data of Korean boys and girls. The data presented are massive and ambitious so as the descriptive analyses. However, the study lacks of focused hypothesis. If mental health and physical activity are the focus of the study as stated in the title, other factors may be treated as confounding factors. Whether physical activity or mental health influence diet or vice versa is unknown. The manuscript should avoid stating the direction of causality. All the diet data are frequency-based, more related to dietary behavior rather than actual intake.
Minor comments:
The manuscript should be more concise. Author had many repeated information in the text and tables. Table 2 may be removed if the survey is an online material. Table 3 and table 4, are they the same table? Any test for the data in these tables? What is fatigue recovery in the table 5? Total column can be removed from Table 5. Table 6 and table 7, provide P value, not intermediate statistics.
Explain more about weight method in data analysis
“The frequency of eating breakfast was found to increase from 10,076 (14.8%) in 2015 to 11,444 (20.0%) in 2019 and to decrease from 32,693 (48.0%) in 2015 to 22,267 (38.8%) in 2019. “ This sentence is confusing.
SES could be confounding factors in logistic regression model. However, how the confounding factors were identified was not stated.
Overall, the paper may be trimmed to test more focused hypotheses that may be novel to the field. Manuscript language and presentation should be improved.
Author Response
I've attached the responses.

Reviewer 2 Report
The manuscript titled 'An investigation on Korean adolescents' dietary consumption: Focused on sociodemographic characteristics, physical health, and mental health' is scientifically sound and very well written. The result has been well explained and discussed. I just have few minor comments:
- From lines 156-172, the authors mentioned the trends of eating breakfast, vegetables, milk intake, and so on. It would be more clear if the authors could also mention whether the increase or decrease in percentages observed was significant or not. That way it would give an idea of the effect of these differences.
- Please check the formatting of the document, such as smaller font for the footnotes of the tables. The discussion title should be on a new line etc.
Author Response
I've attached the responses.

Reviewer 3 Report
Thanks for the opportunity to review the article entitled: An investigation on Korean adolescents' dietary consumption: Focused on sociodemographic characteristics, physical health, and mental health.
There are some comments to this paper.
Abstract
Have no comments for this section
Introduction section
Have no comments for this section
Material and methods
The authors used the survey KYRBWS, which was conducted in 2015, 2017, and 2019. They also describe that this survey included approximately 800 schools each year. It would reinforce the work a lot if they described how this selection is made. Is it a random selection? by strata? Did it include different regions of the country?
The term "dichotomized" to describe the variables related to daily dietary intake is not the most appropriate. Many of these variables have more than answer options. Please changed this term.
Results
In lines 156 to 172, the authors make statements that are not valid since they lack the statistical proof to support them. Some examples are: The frequency of eating breakfast was found to increase from 10,076 (14.8%) in 2015 to 11,444 (20.0%) in 2019 and to decrease from 32,693 (48.0%) in 2015 to 22,267 (38.8%) in 2019. The frequency of vegetable intake also showed a decreasing number of adolescents reporting 'high intake' increased from 298 (0.5%) in 2015 to 437 (0.8%) in 2019. Please add a p-value to support this affirmation or change the redaction.
Do the authors make comparisons of the variables described in Table 5? Would it be interesting to know if there have been modifications of these variables over the years? There were probably changes in the physical and mental health status, and this, in turn, could influence the Dietary intakes.
Tables 7 through 9 may be more informative to put the p-values in place of the x2 values.
In some tables where the p values are presented, it is necessary to correct p-values equal to 0.000. There are no p-values equal to 0.
In the regression analysis, could the authors explain why they chose diet rather than physical health and mental health as the dependent variable?
Odds ratios can sometimes be interpreted as risk. Why did the authors decide to only interpret them as an increment or decrement?
In logistic regressions, the R2 values are shallow. How would the authors reinforce the validity of these results?
It would be interesting to explore the association whit Dietary intakes in multiple logistic regression models.
How the results are represented is a bit confusing; as a suggestion, the authors could rethink the way in which they are represented.
Discussion section
Some papers could be improved in the discussion section.
Song S, Song H. Dietary and Lifestyle Factors Associated with Weight Status among Korean Adolescents from Multicultural Families: Using Data from the 2017–2018 Korea Youth Risk Behavior Surveys. Korean J Community Nutr. 2019 Dec;24(6):465-475. https://doi.org/10.5720/kjcn.2019.24.6.465
Hong, S.A., Peltzer, K. Dietary behaviour, psychological well-being and mental distress among adolescents in Korea. Child Adolesc Psychiatry Ment Health 11, 56 (2017). https://doi.org/10.1186/s13034-017-0194-z
Author Response
I've attached the responses.

Round 2
Reviewer 3 Report
I have no further comments to make, the authors were kind enough to make all the requested modifications. The article improved substantially.
Author Response
We greatly appreciate that you took time out of your busy schedule to review this study. Thank you so much for your positive comments to this work.